# Preparation and Properties of Composite Graphene/Carbon Fiber Pouring Conductive Asphalt Concrete

**DOI:** 10.3390/polym15081864

**Published:** 2023-04-13

**Authors:** Zhenxia Li, Tengteng Guo, Yuanzhao Chen, Yibin Wang, Xiangjie Niu, Deqing Tang, Menghui Hao, Xu Zhao, Jinyuan Liu

**Affiliations:** 1School of Civil Engineering and Communication, North China University of Water Resources and Electric Power, Zhengzhou 450045, China; 2Henan Province Engineering Technology Research Center of Environment Friendly and High-Performance Pavement Materials, Zhengzhou 450045, China; 3Zhengzhou City Key Laboratory of Environmentally Friendly High Performance Road and Bridge Materials, Zhengzhou 450045, China; 4Henan Communications Planning & Design Institute Co., Ltd., Zhengzhou 450046, China

**Keywords:** graphene, carbon fiber, road performance, electrothermal performance, microscopic analysis

## Abstract

To solve the problem of snow on steel bridge areas endangering traffic safety and low road traffic efficiency in winter, conductive gussasphait concrete (CGA) was prepared by mixing conductive phase materials (graphene and carbon fiber) into Gussasphalt (GA). First, through high-temperature rutting test, low-temperature bending test, immersion Marshall test, freeze–thaw splitting test and fatigue test, the high-temperature stability, low-temperature crack resistance, water stability and fatigue performance of CGA with different conductive phase materials were systematically studied. Second, the influence of different content of conductive phase materials on the conductivity of CGA was studied through the electrical resistance test, and the microstructure characteristics were analyzed via SEM. Finally, the electrothermal properties of CGA with different conductive phase materials were studied via heating test and simulated ice-snow melting test. The results showed that the addition of graphene/carbon fiber can significantly improve the high-temperature stability, low-temperature crack resistance, water stability and fatigue performance of CGA. The contact resistance between electrode and specimen can be effectively reduced when the graphite distribution is 600 g/m^2^. The resistivity of 0.3% carbon fiber + 0.5% graphene rutting plate specimen can reach 4.70 Ω·m. Graphene and carbon fiber in asphalt mortar construct a complete conductive network. The heating efficiency of 0.3% carbon fiber + 0.5% graphene rutting plate specimen is 71.4%, and the ice-snow melting efficiency is 28.73%, demonstrating good electrothermal performance and ice-snow melting effect.

## 1. Introduction

Since the mid-1980s, the construction of long-span bridges in China has mushroomed. Due to the advantages of steel box girder orthotropic deck structures, such as lightweight, high strength, good seismic resistance, high efficiency in industrial manufacturing and standardization of construction mode [1], it is widely used in long-span steel bridge structures. However, within the design and use period, the long-span steel bridge completed and put into use in China, the pavement layer has different degrees of micro cracks and wave passage phenomena [2], and a few bridge deck pavement layers even show local congestion and rutting damage. Gussasphalt (GA) is widely used in steel bridge deck pavement structures because of its good impermeability, aging resistance, fatigue resistance and good deformation compliance. At high temperatures (220–240 °C), GA has better fluidity than ordinary asphalt concrete, which gives it the characteristics of self-compacting molding. When paving the pavement, it does not need rolling, and the whole paving process can be completed through leveling machinery [3]. In winter, the snow and ice crossings on the bridge deck not only reduce traffic efficiency but also bring great threats and challenges to traffic safety. Therefore, it is essential to remove the snow on the road in time to ensure the normal operation of the traffic. Usually, the methods of removing snow and ice include spreading chlorine salt chemical snow melting agent, artificial snow cleaning, and mechanical snow removal [4]. Although this can achieve the purpose of removing snow and ice, chloride ions and chemical agents will penetrate the internal structure of the bridge deck and cause corrosion of steel bars, and also pollute the environment. Moreover, in the working process of the machine, aggregates on the road surface can easily be peeled off, resulting in road damage. In this case, many scholars have turned their attention to the study of conductive asphalt concrete. By adding conductive phase materials to the asphalt mixture, under a continuous and stable input voltage, the current thermal effect (Joule’ s law) is used to achieve the warming and melting of ice and snow on bridge decks [5]. The conductive layer is set on the upper layer, and the insulation layer is set at the bottom to avoid heat loss and improve the efficiency of melting ice and snow [6].

Conductive gussasphalt concrete (CGA), which combines the advantages of both cast asphalt concrete and conductive asphalt concrete, can meet the requirements of long-span steel bridge deck pavement. At the same time, CGA can meet the requirements of road snow melting and ice melting and ensure normal operation of the traffic system during snow and ice conditions. During the snow-removal process, there is no need to interrupt traffic and the heating effect remains stable [7,8]. Moreover, this does not cause pollution to the environment, which meets the environmental protection requirements advocated by China. Therefore, the research on CGA suitable for steel bridge deck pavement holds important reference value and practical significance for improving road traffic efficiency and reducing ice- and snow-related disasters.

When graphite is used as the conductive phase material to prepare conductive asphalt concrete, the conductive performance of asphalt concrete can be significantly improved when the content of graphite is large, and its conductive performance is positively correlated with the content of graphite. Although the incorporation of graphite into asphalt concrete can greatly improve its conductive performance, due to the lubrication effect of graphite itself and easy dissociation [9,10], the shear resistance of asphalt concrete is reduced, thereby reducing its road performance. Graphene has excellent electrical conductivity, high-temperature resistance, corrosion resistance and good compatibility with asphalt concrete [11]. When the content of graphene is 0.5%, the prepared rut plate specimens have good electrical conductivity [12], with high temperature performance and fatigue resistance also improving to some extent. The pavement performance of asphalt concrete can be significantly improved by adding carbon fibers with different contents and sizes to the asphalt mixture. When the length is 3 cm and the content is 0.25% carbon fiber (mass fraction of aggregate) to the asphalt mixture [13], the high-temperature stability and fatigue resistance of the mixture are greatly improved. Compared with the mixture without carbon fiber, its anti-fatigue performance is greatly improved, and the tensile stiffness modulus increases by 24.5%. The influence of carbon fiber with different sizes on its pressure sensitivity characteristics is obvious [14]. Although the single addition of carbon fiber conductive phase material in asphalt concrete can significantly improve the pavement performance of asphalt concrete, the improvement effect on its electrical conductivity is poor [15]. The resistivity of specimens with 25% graphite (asphalt mass fraction) and 0.4% carbon fiber (aggregate mass fraction) was significantly reduced [16,17]. Carbon fiber greatly improved the toughness and crack resistance of concrete and formed a three-dimensional network structure inside the concrete to form a complete conductive network. At the nanometer level, one should not readily discount the effects of Stone–Wales defect on the electrical properties of graphene. In the presence of the Stone–Wales defect, it has been reported that the current flowing through the graphene varies with the number and location of the Stone–Wales defect; the number and location of these defects can affect the density of charge carriers and pathway of the carriers [18]. More importantly, these defects also affect the mechanical performance of the graphene [19]. Stone–Wales defects could interact and influence the strain energy to fracture of these nanoparticles, but the extent of the effects depends on the orientation of the defect arrangement in the direction of the load acting on the nanoparticle and on the mode of deformation, such as tensile and torsion [19,20]. However, with the continuous increase in graphite content, the resistance of the specimen decreased slowly and gradually maintained a certain value. At this time, the conductive performance of asphalt mixture reached the percolation threshold. In the subsequent ice and snow melting test, the specimen showed good electrical and thermal properties.

In addition to carbon fiber, steel fiber and steel cashmere fiber (SWF) also have good electrical conductivity and mechanical properties. Asphalt concrete mixed with graphite and steel fiber conductive phase materials [21] has better high-temperature stability, but its low-temperature crack resistance is reduced to a certain extent, and its road performance is still better than that of ordinary asphalt concrete. The asphalt concrete with SWF as conductive phase material, adding 0.2% SWF (mass fraction of aggregate) can meet the heating effect and strength requirements [22]. The corrosion and passivation of SWF will affect the sustainability of asphalt concrete due to the external environment. However, the sustainability of asphalt concrete is essentially unaffected by low SWF content [23]. Small diameter SWF has a better improvement effect on the strength of asphalt concrete. At the same time, the self-healing ability of asphalt concrete will be enhanced with an increase in temperature in the heating process. Steel slag is used to replace aggregate and conductive phase materials to prepare conductive asphalt concrete, and a small amount of graphene and carbon fiber is added as auxiliary conductive media [24,25]. Compared with basalt asphalt concrete, the dynamic stability of steel slag conductive asphalt concrete is increased by 10.3%. Steel slag can significantly improve the high-temperature stability of asphalt concrete, but the improvement effect of water stability and fatigue resistance is not obvious. In the temperature-rising and snow-melting test, the input power was 520 W/m^2^, and the temperature of the rutting plate specimen increased by 16.7 °C after 30 min of power supply, indicating that the steel slag conductive asphalt concrete had a good snow-melting effect.

In summary, although single-doped carbon fiber significantly enhances the pavement performance of asphalt concrete, it has poor electrical conductivity. Graphite can significantly improve the electrical conductivity of concrete at high content, but the road performance decreases in varying degrees. Asphalt concrete with low graphene content has good electrical conductivity. At the same time, the high temperature performance and fatigue resistance of asphalt concrete are improved to some extent, but the improvement effect on low temperature performance and water stability is not obvious. The passivation of metal conductive phase materials will lead to an increase of resistance, and the sustainability of the gradual decline of electrical conductivity is poor. At the same time, it also has a negative impact on the low-temperature performance and water stability of conductive asphalt concrete. If the characteristics of conductive phase materials are combined, it can be predicted that they will have a profound impact on the development of conductive asphalt concrete. At present, there is little research on the performance of graphene/carbon fiber conductive asphalt concrete. Therefore, in this paper, graphene/carbon fiber composite is used to prepare CGA, which makes up for the shortcomings of single-doped conductive phase materials. The road performance and electrothermal performance of CGA are systematically studied.

## 2. Research Content

On the premise of summarizing the research progress of conductive asphalt concrete at home and abroad, combined with the advantages of conductive phase materials, this paper systematically studies the graphene/carbon fiber cast conductive asphalt concrete through a large number of indoor tests. The main research contents include the following aspects:

(1) Mix design of cast conductive asphalt concrete

According to the specification [26], the pouring asphalt concrete adopts GA-10 gradation, and the GA-10 mix ratio is designed and optimized. Through the fluidity test, penetration and penetration increment test, the optimum content of Sasobit in GA-10 and the optimum oil–stone ratio are determined. On this basis, it is recommended to prepare CGA-10 by mixing 0.3% carbon fiber + (0.4%, 0.5%, 0.6%) graphene conductive phase materials. The effects of different oil–stone ratios on the fluidity and plastic deformation resistance of CGA-10 were studied by fluidity test, penetration and penetration increment test, and the optimal oil–stone ratio was determined.

(2) Research on road performance

Through high-temperature rutting test, low-temperature bending beam test, freeze–thaw splitting test, immersion Marshall test and fatigue test, the influence of conductive phase material content on high-temperature stability, low-temperature crack resistance, water stability and fatigue performance of CGA-10 was studied, and the road performance of CGA-10 was comprehensively evaluated.

(3) Conductivity study

First, the influence of graphite spreading amount on contact resistance between electrode and specimen was studied by resistance test, and the optimal graphite spreading amount was recommended. Second, through the penetration and rutting plate specimen resistance test, the influence of different conductive phase material content and electrode layout scheme on the electrical conductivity is studied, the optimal electrode layout scheme is determined, and the content of conductive phase material is optimized. Finally, the conductive mechanism of CGA-10 was further explained from a microscopic point of view by combining the existing conductive theory of composite materials and scanning electron microscopy.

(4) Study on electrothermal performance

First, the penetration and the thermal conductivity of the rut board specimen were measured to study the influence of different conductive phase material content on its thermal conductivity. The electric heating test and simulated melting ice and snow test were then carried out to analyze the influence of different conductive phase material content on the electrothermal performance of CGA-10 and determine the optimal content of conductive phase material in CGA-10.

## 3. Raw Material

### 3.1. Asphalt

In this experiment, SBS-modified asphalt produced by Zhengzhou Municipal Engineering Corporation is selected, and its main technical indicators are tested according to the specification [27]. The test results are shown in Table 1.

### 3.2. Sasobit

Sasobit (sand rope) has the function of improving asphalt fluidity and aging resistance, so the test uses Sasobit asphalt modifier produced by DuPont Co., Ltd. (Shanghai, China), the main technical indicators are shown in Table 2.

### 3.3. Aggregate

The aggregate and mineral powder produced by Zhengzhou Zhengfa Municipal Co., Ltd. (Zhengzhou, China). was selected in the experiment. The aggregate was composed of limestone rolling crushed stone, and the mineral powder was obtained after limestone grinding. The main technical indexes were tested according to the specification [28].

1.Coarse aggregate

In the asphalt mixture, coarse aggregate can mainly play the role of skeleton inlay and extrusion, which should have good impact resistance and wear resistance. The particle sizes of coarse aggregate are 3–5 mm and 5–10 mm, and the related indexes should meet the requirements of the specification. The main technical indexes are shown in Table 3 and Table 4.

2.Fine aggregate

The geometric characteristics of fine aggregate have an important influence on the performance of asphalt concrete. The basic indexes are tested according to the specification [28]. The main technical indexes are shown in Table 5.

### 3.4. Mineral Powder

The mineral powder used in this paper is obtained by grinding limestone. According to the test method of the specification [28], the basic indexes of mineral powder are tested. The main technical indexes are shown in Table 6.

It can be seen from Table 6 that the ore powder used in the test meets the requirements of the specification.

### 3.5. Graphene

Graphene produced by carbon-rich enterprises was selected for the experiment. Graphene is shown in Figure 1, and the microstructure is shown in Figure 2. The main technical indicators are shown in Table 7.

It can be seen from Figure 1 and Figure 2 and Table 7 that graphene is a powder solid in the macroscopic state, the overall color is blackish to darker. When magnified by 5000 times and 10,000 times, graphene is a multilayer sheet-like stacking structure. Because graphene has a large specific surface area and has a strong adsorption effect on asphalt, dispersing it into an asphalt mixture can increase the adhesion between asphalt and mineral aggregate.

### 3.6. Carbon Fiber

Relevant studies have shown [29] that the dispersion of long carbon fiber in asphalt mixture is poor, and easily agglomerates, resulting in an uneven mixture. Although the dispersion of short carbon fiber is uniform, it is difficult to build a conductive network to affect the conductivity. When the length of carbon fiber is 9 mm, it has a good dispersion effect and compatibility in the asphalt mixture and can prepare a mixture with more uniform dispersion of conductive phase materials. Therefore, 9 mm carbon fiber produced by Dongli Co., Ltd. (Shanghai, China) was selected in this experiment, and the main technical indexes are shown in Table 8.

## 4. Mix Proportion Design of Casting Conductive Asphalt Concrete

The mixed design method of cast conductive asphalt concrete is different from ordinary asphalt concrete. It is mainly based on ‘Technical specification for design and construction of highway steel bridge deck pavement’ (JTG/T3364-02-2019) [26] (hereinafter referred to as steel bridge deck pavement specification). Its main technical indicators are fluidity 5–20 s, penetration 1–4 mm, and penetration increment ≤ 0.4 mm. The low-temperature performance verification was carried out after determining the optimal oil–stone ratio of GA-10, and the specific mix design process is shown in Figure 3.

According to the design process of CGA-10 mixture ratio shown in Figure 3, the optimum dosage of Sasobit was determined by the Liuer fluidity test. Combined with the fluidity test results, the penetration and penetration increment test were carried out to determine the optimum oil–stone ratio of GA-10. On this basis, CGA-10 was prepared by mixing graphene and carbon fiber, and the optimal oil–stone ratio of CGA-10 was determined through the flow test, penetration and penetration increment test.

The aggregates are divided into three grades: 1# (5–10 mm), 2# (3–5 mm) and 3# (0–3 mm). The screening results are shown in Table 9.

According to the specification of steel bridge deck pavement [26], the pouring asphalt concrete adopts GA-10 gradation. The proportion of aggregate and mineral powder of each grade is adjusted via EXCEL table as 5–10 mm: 3–5 mm: 0–3 mm: mineral powder = 27: 22: 26: 25. The final composite gradation of the mixture is shown in Table 10, and the composite gradation curve is shown in Figure 4.

### 4.1. Determination of Sasobit Content and Optimum Oil–Stone Ratio of Cast Asphalt Concrete

Appropriate fluidity is the key to ensuring the self-compacting molding of cast asphalt concrete. The incorporation of Sasobit can improve the fluidity and aging resistance of the asphalt mixture. Combined with the GA practical engineering of rigid bridge deck pavement in China [3], it is estimated that the median oil-aggregate ratio of GA-10 is 8.5%, floating up and down 0.2%, taking 8.1%, 8.3%, 8.5%, 8.7% and 8.9% as five oil-aggregate ratios, and a Sasobit dosage of 3% as the median (asphalt mass fraction), floating up and down 1.5%, taking 0.0%, 1.5%, 3.0%, 4.5% and 6.0% as five dosages.

#### 4.1.1. Fluidity Test of Cast Asphalt Concrete

Flowability is an important index to evaluate the workability and high temperature performance of GA-10 construction. When the flowability of asphalt mixture is within 5–20 s, it shows that it has good flowability and high temperature performance. According to the specification of steel bridge deck pavement [26], the preparation process of GA-10 is first, the heated coarse and fine aggregates are poured into the stirring pot for 15 s. Sasobit and asphalt are added into the mixing pot and stirred for 90 s. To ensure that GA-10 has good fluidity, the final addition of ore powder needs 40 min stirring. The test results are shown in Figure 5.

It can be seen from Figure 5 that:(1)The results of GA-10 liquidity test with oil–stone ratios of 8.1% and 8.9% did not meet the requirements of the specification for 5–20 s. When the oil–stone ratio is 8.3%, 8.5% and 8.7%, the fluidity is 16.3 s, 11.7 s and 8.4 s, respectively, which meet the requirements of the specification. The mixture has good fluidity.(2)When the content of Sasobit in the asphalt mixture increases from 0 to 3.0%, the fluidity of GA-10 with 8.3% oil-aggregate ratio increases by 18.2%, and when the content of Sasobit continues to increase from 3.0 to 6.0%, the fluidity increases by 5.5%. Similarly, the liquidity of GA-10 with the oil–stone ratio of 8.5% increases by 25.5% and 7.7%, and the liquidity of GA-10 with the oil–stone ratio of 8.7% increases by 29.4% and 7.1%. It can be found that when the content of Sasobit is less than 3.0%, the increase rate of mixture fluidity is the fastest. When the content exceeds 3.0%, the growth rate begins to slow down, and the improvement effect of Sasobit on mixture fluidity is weakened.(3)A Sasobit content of 3.0% can significantly improve the fluidity of GA-10, thereby reducing the difficulty of mixture paving in construction process and improving the construction efficiency. However, excessive Sasobit content in the mixture will lead to a decrease in the bearing capacity and fatigue resistance of the asphalt pavement. Therefore, the optimal Sasobit content in GA-10 is determined to be 3.0%.

#### 4.1.2. Poured Asphalt Concrete Penetration and Penetration Increment Test

The penetration and penetration increment can characterize the plastic deformation resistance of GA-10 in a high temperature environment. In this test, the LHGR-I asphalt concrete penetration meter was used. The main instruments include:(1)Weight: 10~50 kg.(2)Dial indicator: range 0~10 mm, accuracy 0.01 mm.(3)Penetration rod: diameter 25.2 mm, mass 2.5 kg, cross-sectional area 5 cm^2^.(4)Constant temperature water bath: 60 °C, precision 0.1 °C.(5)Penetration test: 70.7 × 70.7 × (70.7 ± 1) mm.

When the initial pressure is not added, the weight is only dependent on the gravity of the penetrating rod itself (the load is 24.5 N). When the final pressure is loaded, four weights (the load is 490.5 N) are loaded. The penetration degree of the specimen is the sinking depth of the penetrating rod after an initial pressure of 10 min and a final pressure of 30 min. The penetration increment is the difference between the readings of the dial gauge after the final pressure of 30 min and 60 min. The penetration and its increment of three specimens in each group were measured and the average value was calculated. The test process is shown in Figure 6, and the test results are shown in Figure 7.

It can be seen from Figure 7 that when the oil–stone ratio of GA-10 is 8.3% and 8.7%, the test results do not meet the penetration of 1–4 mm required by the specification. When the oil–stone ratio is 8.3%, 8.5% and 8.7%, with the increase in oil–stone ratio in GA-10, the penetration increment test results are consistent with the penetration test results. When the oil–stone ratio in GA-10 is 8.7%, the penetration increment that does not meet the specification requirements is ≤ 0.4 mm. Therefore, the optimum oil–stone ratio of GA-10 was 8.5%. According to the specification of steel bridge deck pavement [26], the low temperature performance is verified by low-temperature bending test. The test results are shown in Table 11.

It can be seen from Table 11 that when the GA-10 oil–stone ratio is 8.5%, the low-temperature bending test proves that the optimized GA-10 mix design meets the specification requirements.

### 4.2. Conductive Phase Material Content and Its Dispersion Method

1.Content of conductive phase materials

The related research on conductive asphalt concrete shows that when the graphene content in the asphalt mixture is 0.5% (asphalt mass fraction), the electrical conductivity is significantly improved [30]. When the content of carbon fiber in asphalt mixture is 0.3% (mass fraction of aggregate), the dispersion of carbon fiber in asphalt mixture is good, but the conductivity is poor, and there is still a lot of room for improvement. When the content of carbon fiber continues to increase, it is easy to produce cohesive agglomeration due to its adsorption on asphalt, which leads to the decrease of dispersion in asphalt mixture and reduces the road performance and conductivity of conductive asphalt concrete to a certain extent. Therefore, in this experiment, the composite doping method was adopted. The three dosages of graphene were 0.4%, 0.5% and 0.6%, respectively, and the dosage of carbon fiber was 0.3%.
2.Dispersion method of conductive phase materials

① Carbon fiber dispersion

Due to the strong adsorption of carbon fiber on asphalt, the agglomeration phenomenon occurs under the cementation of asphalt, resulting in a significant decrease in dispersion. Therefore, in the preparation of conductive asphalt mixture, the order of incorporation of carbon fiber seriously affects its dispersion effect and the uniformity of conductive asphalt mixture. To make the conductive phase materials in CGA-10 disperse uniformly, the stirring time after adding conductive phase materials is appropriately prolonged. According to the specification of steel bridge deck pavement [26], the final addition of ore powder requires 40 min stirring to ensure good fluidity of the GA-10. In total, three schemes are used in the test, as shown in Figure 8.

Scheme 1: Adding carbon fiber after mixing the asphalt and aggregate evenly will not cause the agglomeration phenomenon to occur in the mixing process. Instead, the carbon fiber and aggregate have been mixed evenly after mixing. At this time, adding asphalt will bond the dispersed carbon fiber and aggregate together to obtain the asphalt mixture with good dispersion effect, which can form a complete and uniform conductive network. Scheme 2 and Scheme 3: The carbon fiber is first added to mix evenly with the aggregate, and then asphalt is added, or the carbon fiber is directly dispersed in the asphalt. Both methods produce serious agglomeration. This is mainly due to the concave and convex stripes on the surface of the bundle carbon fiber, resulting in a small contact area between each other, resulting in stress concentration. At the same time, liquid asphalt is different from aggregate and cannot form a solid skeleton structure. Finally, due to the bonding and stress effect in the mixing process, serious agglomeration occurs.

According to Scheme 1, the carbon fiber and aggregate were mixed evenly, and then asphalt was added to prepare the asphalt mixture with good carbon fiber dispersion. The test process is shown in Figure 9, and this scheme is used in subsequent tests.

② Graphene dispersion

Because graphene has a large specific surface area and van der Waals force between layers, it is difficult to disperse graphene directly in the asphalt mixture, which limits its practical application. Therefore, in this experiment, graphene was dispersed into the asphalt in advance by high-speed shear. In the experiment, the shear rate was 5500 r/min, the temperature was 160 °C, and the shear time was 30 min. The test process is shown in Figure 10.

The microstructure of asphalt after high-speed shear dispersion was analyzed by scanning electron microscopy, as shown in Figure 11.

It can be seen from Figure 11 that when the magnification is 500 and 1000 times, it can be clearly observed that graphene is uniformly dispersed in the asphalt, indicating that the use of high-speed shear can effectively improve the dispersion of graphene in asphalt.

### 4.3. Determination of Optimum Oil–Stone Ratio of Cast Conductive Asphalt Concrete

Due to the strong adsorption effect of graphene and carbon fiber on asphalt, if the optimum asphalt–aggregate ratio is maintained at 8.5%, the mixture will become viscous and lose the original fluidity and plastic deformation resistance, which cannot meet the technical indexes of fluidity, penetration and penetration increment required by the steel bridge deck pavement specification. Therefore, it is necessary to appropriately increase the oil–stone ratio based on 8.5%. When estimating the oil–stone ratio of CGA-10, the influence of conductive phase materials on the oil–stone ratio is considered [31]. The median value of the oil–stone ratio of CGA-10 is estimated to be 9.8%, and it fluctuates up and down by 0.2%, namely, 9.4%, 9.6%, 9.8%, 10.0% and 10.2%.

#### 4.3.1. Fluidity Test of Cast Conductive Asphalt Concrete

The incorporation of conductive phase materials has a significant indigenous effect on the fluidity of CGA-10. The three-time fluidity of each group of mixtures is measured and the mean value is calculated. The flow test results are shown in Figure 12.

It can be seen from Figure 12 that when the oil–stone ratio of CGA-10 is 9.4%, the fluidity of the three groups of conductive phase materials is 20.4 s, 21.5 s and 22.3 s, respectively, which cannot meet the requirements of the steel bridge deck pavement specification for 5–20 s. Currently, the mixture is relatively viscous and has poor fluidity. When the oil–stone ratio is 10.2%, the fluidity of the three groups of conductive phase materials is 3.0 s, 3.7 s and 4.3 s, respectively, which also does not meet the requirements of the steel bridge deck pavement specification. When the asphalt-aggregate ratio is 9.6%, 9.8% and 10.0%, the test results of CGA-10 asphalt mixture meet the requirements of steel bridge deck pavement specification for 5–20 s with good fluidity under the three groups of conductive phase materials. Therefore, in the three groups of conductive phase material content, the preliminary determination of CGA-10 petroleum ratio range is 9.8%, 10.0%, 10.2%.

#### 4.3.2. Incremental Test of Penetration and Penetration of Cast Conductive Asphalt Concrete

Three penetration specimens were prepared under each dosage, and the average test results were taken. Under the oil–stone ratio of 9.6%, 9.8% and 10.0%, the penetration test results of CGA-10 are shown in Figure 13 and the incremental penetration test results are shown in Figure 14.

From Figure 13 and Figure 14, it can be seen that when the oil–stone ratio is 9.6%, the penetration of CGA-10 under the three groups of conductive phase materials is 0.93 mm, 0.81 mm and 0.72 mm, respectively, which do not meet the requirements of steel bridge deck pavement specifications. The penetration of 1–4 mm. When the oil ratio is 10.0%, the penetration test results also do not meet the requirements of the steel bridge deck pavement specification. When the oil–stone ratios were 9.6%, 9.8% and 10.0%, with the increase of oil–stone ratio in CGA-10, the results of penetration increment test were consistent with those of penetration test. When the oil–stone ratio is 9.8%, the penetration increments of CGA-10 with three conductive phase materials are 0.23 mm, 0.21 mm and 0.19 mm, respectively, which meet the requirements of steel bridge deck pavement specifications. The penetration increment is less than 0.4 mm. Therefore, with the addition of 0.3% carbon fiber + (0.4%, 0.5%, 0.6%) graphene in three groups, the optimum asphalt-aggregate ratio of CGA-10 was determined to be 9.8%, and the mixture had good fluidity and plastic deformation resistance.

## 5. Study on Pavement Performance of Pouring Conductive Asphalt Concrete

### 5.1. High-Temperature Stability

According to the specification of steel bridge deck pavement, the dynamic stability requirements of CGA-10 rut plate specimens are not less than 300 times/mm. By comparing the rutting test results of CGA-10 and GA-10, the influence of conductive phase material content on the high-temperature stability of asphalt mixture is analyzed. The test results are shown in Table 12.

It can be seen from Table 12 that compared with the GA-10 rutting test results, the dynamic stability of the specimen doped with 0.3% carbon fiber is increased by 13.9%, indicating that the incorporation of carbon fiber into asphalt mixture can significantly improve its high temperature stability. It is mainly due to the good mechanical properties of carbon fiber and the strong adsorption of asphalt so that it can be better cemented and wrapped with asphalt and minerals. With the decrease in free asphalt and the increase in structural asphalt, the shear resistance and high temperature stability of CGA-10 are significantly improved. Compared with the rutting plate specimens doped with 0.3% carbon fiber, the dynamic stability of CGA-10 with conductive phase materials increases by 12.3%, 17.6% and 22.9% respectively with the increase of graphene content, indicating that the incorporation of graphene can further improve the high-temperature stability of CGA-10. It is mainly because graphene can effectively combine with partial free asphalt, thereby increasing the proportion of structural asphalt, enhancing the plastic deformation resistance of CGA-10 and improving the high-temperature stability.

### 5.2. Low-Temperature Crack Resistance

The low-temperature bending test is to apply concentrated load to the mid-span of the beam to make it fracture. The rut plate is cut out of the beam specimen by the cutting machine and placed in the incubator at −10 °C ± 0.5 °C for 5 h. The spacing of the test machine fulcrum is adjusted to 200 ± 0.5 mm, and the mid-span loading is carried out at a rate of 50 mm/min. The test results are shown in Table 13.

It can be seen from Table 13 that the flexural tensile strength, maximum flexural tensile strain and flexural stiffness modulus of the trabecular specimens with 0.3% carbon fiber were increased by 36.7%, 27.2% and 7.4%, respectively, compared with the GA-10 test results, indicating that the incorporation of carbon fiber into the asphalt mixture can significantly improve its low-temperature crack resistance, mainly because carbon fiber has good mechanical properties, can resist CGA-10 cracking and crack development, improve its toughness and self-healing ability, and change from low temperature brittle failure to ductile failure, thereby enhancing its low temperature performance. Compared with the test results of trabecular specimens with 0.3% carbon fiber, with the increase of graphene content, the flexural tensile strength of CGA-10 trabecular specimens with conductive phase materials decreased by 5.4%, 10.0% and 16.4%, respectively, the maximum flexural tensile strain decreased by 5.8%, 8.0% and 9.9%, respectively, and the flexural stiffness modulus decreased by 0.7%, 2.2% and 4.9%, respectively. It shows that the incorporation of graphene in CGA-10 has a certain negative impact on its low temperature crack resistance, mainly because graphene adsorbs some free asphalt, which increases the proportion of structural asphalt. Furthermore, the brittleness of asphalt also increases, resulting in a decrease in low-temperature crack resistance.

### 5.3. Water Stability

Due to the external freeze–thaw environment and the repeated action of variable driving load, the dynamic water pressure is continuously generated inside the structure. Under the action of dynamic water pressure, the asphalt attached to the surface of the mineral aggregate can migrate. When the asphalt film in the asphalt mixture is thin, the water infiltrated into the structure can penetrate the asphalt film, so that the asphalt is separated from the surface of the mineral aggregate and the mixture is loose, which has a serious impact on its strength and durability.

#### 5.3.1. Immersion Marshall Test

The results of immersion Marshall test are shown in Table 14.

It can be seen from Table 14 that the residual stability of CGA-10 mixed with conductive phase material is reduced by 0.7%, 0.9% and 1.9%, respectively, with the increase of graphene content compared with Marshall specimen doped with 0.3% carbon fiber, and the decrease is very small. It shows that the water stability of CGA-10 mixed with graphene is slightly reduced, mainly due to the hydrophobicity of graphene and the adsorption of free asphalt, which leads to the decrease of asphalt adhesion, and then reduces the water stability of CGA-10. Compared with GA-10, the residual stability of Marshall specimen doped with 0.3% carbon fiber increased by 5.2%, indicating that the incorporation of carbon fiber into asphalt mixture can improve its water stability. The main reason is that carbon fiber can form a three-dimensional network structure in asphalt mixture, which strengthens the overall stability of CGA-10, effectively prevents water from entering the interior of asphalt concrete, and thus improves its water stability.

#### 5.3.2. Freeze–Thaw Splitting Test

A total of eight standard Marshall specimens were prepared in each group. The splitting strength before and after freezing and thawing was measured by the stability instrument. The test results are shown in Table 15.

It can be seen from Table 15 that the TSR values of Marshall specimens are greater than those of GA-10 without conductive phase materials, regardless of the single- or multiple-doped conductive phase materials in CGA-10. The influence of the increase in the content of conductive phase material in CGA-10 on its TSR value is consistent with that of the immersion Marshall test. With the increase of graphene content in CGA-10, the TSR value of Marshall specimens decreased by 0.7% and 0.8%, respectively, indicating that with the increase of graphene content, the water stability of CGA-10 decreased slightly. Overall, compared with GA-10 without conductive phase material, the water stability of CGA-10 with graphene and carbon fiber is better.

### 5.4. Fatigue Property

The loading frequency was adjusted to 10 Hz with 400 με strain level. The three specimens were numbered and put into the curing box, the temperature in the curing box was controlled to 15 °C, and the indoor temperature was adjusted to 15 °C. The test results are averaged. The test results are shown in Table 16.

It can be seen from Table 16 that compared with GA-10, the average fatigue number of specimens with 0.3% carbon fiber increased by 6.2%, indicating that the incorporation of carbon fiber into asphalt mixture can improve the fatigue resistance of mixture to a certain extent. Compared with the fatigue specimen doped with 0.3% carbon fiber alone, the average fatigue times of CGA-10 doped with conductive phase materials increased by 4.5%, 5.8% and 7.7%, respectively, with the increase of graphene content, indicating that the incorporation of graphene into asphalt mixture can improve the fatigue performance of mixture to a certain extent.

## 6. Study on Electrothermal Performance of Cast Conductive Asphalt Concrete

The good electrothermal performance of conductive asphalt concrete is key to realizing snow melting and ice melting under the condition of electricity. This chapter mainly analyzes the influence of different conductive phase material content on the electrothermal performance of CGA-10 through the rutting plate resistance test, the electric heating test and the simulated snow melting test and determines the optimal conductive phase material content.

### 6.1. Determination of Graphite Distribution on Contact Surface between Electrode and Specimen

Due to the void and unevenness on the surface of the formed rutting plate specimen, the contact resistance between the specimen and the electrode increases. In the resistance measurement, to reduce the contact resistance and ensure a greater degree of adhesion between the electrode and the specimen, graphite powder is selected as the intermediate conductive medium. The DDG-A conductive paste was uniformly smeared on the surface of the specimen contacted with the electrode in advance, and then the graphite was evenly distributed between the specimen and the electrode, so as to prevent the graphite particles from falling off during the test leading to large errors in the results. When the external electrode scheme is adopted, the gap between the contact surfaces is shown in Figure 15. The test process is shown in Figure 16, and the test results are shown in Figure 17.

### 6.2. Resistance Test of Rutting Plate

Three groups of CGA-10 rut plate specimens were prepared by adding 0.3% carbon fiber + 0.4% graphene, 0.3% carbon fiber + 0.5% graphene and 0.3% carbon fiber + 0.6% graphene, respectively. Three specimens were prepared under each dosage. The external copper electrode test method was used. The average test results were taken, and the resistivity was calculated, as shown in Table 17.

It can be seen from Table 17 that with the increase of graphene doping, the resistivity of the specimen gradually decreases. When the conductive phase material content was 0.3% carbon fiber + 0.4% graphene, the resistivity of the specimen was the largest, which was 6.81 Ω·m. When the graphene content increased from 0.4% to 0.5%, the resistivity of the specimen decreased by 2.11 Ω·m. When the graphene content continued to increase, the resistivity of the specimen changed little, indicating that the conductivity of CGA-10 reached the percolation threshold. With the increase of the conductive phase material content, the resistivity of the specimen did not change significantly. When the conductive phase materials are 0.3% carbon fiber + 0.5% graphene and 0.3% carbon fiber + 0.6% graphene, the resistivity of rut plate specimens are 4.70 Ω·m and 4.51 Ω·m, respectively, indicating that it has good electrical conductivity and can be used for the next electrothermal performance research.

### 6.3. Mechanism Analysis of Graphene and Carbon-Fiber-Reinforced Conductivity

The microstructure of CGA-10 was analyzed by scanning electron microscopy, as shown in Figure 18.

It can be seen from Figure 18 that when the magnification is 300 times, the carbon fibers are overlapped with each other in a staggered distribution, and are well combined and wrapped with mineral materials and asphalt. The carbon fibers and graphene uniformly distributed in asphalt mortar cooperate to construct a complete conductive network. When the magnification is 1000 times, it is found that in CGA-10, conductive phase materials, mineral materials and asphalt are closely bonded and interlocked. Since carbon fiber has good mechanical properties and electrical conductivity, it can inhibit the development of cracks in asphalt concrete and improve the flexibility and electrical conductivity of CGA-10. In CGA-10, carbon fibers are interlaced and overlapped with each other, forming a good and stable conductive network with uniformly distributed graphene. According to the percolation theory, when the content of carbon fiber and graphene is small, the matrix cannot form a good conductive percolation network, and the resistance of the mixture decreases but still does not have the conductive ability. When the content of carbon fiber and graphene reaches the percolation threshold, the resistance of the mixture decreases sharply, and the conductive percolation network is formed. Then with the continuous increase of the content of graphene and carbon fiber in CGA-10, the conductive phase materials can be more contacted, and the resistivity change gradually tends to be gentle.

### 6.4. Temperature Rise Test of Rutting Plate Specimen

The rutting plate specimens with 0.3% carbon fiber + 0.5% graphene and 0.3% carbon fiber + 0.6% graphene were subjected to the electrified heating test, and the real-time temperature value of the specimen surface was measured by a digital temperature sensor. The temperature values at each measuring point of the two groups of specimens were compared and analyzed in the same electrified time, and the heating efficiency was calculated. A total of six temperature measuring points are selected, which are four measuring points at the position of 5 cm from the edge on the diagonal line of the upper layer, and two measuring points at the center position of the upper and lower layers. The average temperature at the five positions of points A, B, C, D and E is the upper surface temperature of the rutting plate specimen, and point F is the bottom temperature of the specimen. The distribution of specific temperature measuring points is shown in Figure 19.

When 0.3% carbon fiber + 0.5% graphene and 0.3% carbon fiber + 0.6% graphene were mixed, the mass of the prepared rut plate specimens was 10.97 kg and 10.93 kg, respectively. According to the size and mass of the formed rut plate specimens, the density of CGA-10 with two groups of conductive phase materials was calculated to be 2438 kg/m^3^ and 2429 kg/m^3^, respectively. According to the relevant experimental research data [32], the electrothermal parameters of the relevant materials are shown in Table 18.

The specific heat capacity of CGA-10 was calculated according to the composite Formula (1):(1)$$c_{\alpha }=\frac{\rho_{GR}}{\rho_{\alpha }}\times v_{GR}\times c_{GR}+\frac{\rho_{CF}}{\rho_{\alpha }}\times v_{CF}\times c_{CF}+\frac{\rho_{AC}}{\rho_{\alpha }}\times (1-v_{CF}-v_{GR})\times c_{AC}$$

The specific heat capacity, density and volume fraction of carbon fiber.

The specific heat capacity, density and volume fraction of, -graphene.

Through the calculation of the composite material Formula 2, it can be seen that:

The specific heat capacity of CGA-10 is 820.3 J/(kg·K) when 0.3% carbon fiber + 0.5% graphene is added.

The specific heat capacity of CGA-10 is 827.5 J/(kg·K) when 0.3% carbon fiber + 0.6% graphene is added.

The specific heat capacity of CGA-10 under two groups of different conductive phase materials has little difference, which is not much different from that of ordinary modified asphalt concrete 831.2 J (kg·K), mainly due to the small specific heat capacity of graphene and carbon fiber itself. At the same time, the volume fraction of conductive phase material in CGA-10 is small compared with that of mineral aggregate, so the addition of conductive phase material has little effect on its specific heat capacity.

According to the total energy output from the power supply, the temperature change value of the specimen, the specific heat capacity and quality of each rut plate specimen, the heat storage capacity of the specimen can be calculated, and the heating efficiency of CGA-10 can be obtained. The output voltage of single-phase AC220V AC STG voltage regulator is set to 50 V, and the electrode is arranged externally. The relative humidity is 70%, and the test temperature is 15.3 °C at room temperature. Under the same power-on time, the electrothermal performance is evaluated according to the surface temperature change of CGA-10 rutting plate specimen under two groups of different conductive phase materials.

The total test time is 2 h, and then record the number of digital thermometers every 10 min. It is found that the heating rate is slow in the electric heating test of the rut plate specimen. Therefore, the temperature change of the specimen is recorded every 20 min after 1 h. The test process is shown in Figure 20. The test results are shown in Table 19 and Table 20.

According to Table 19 and Table 20, the linear fitting between the surface temperature of the rutting plate specimen and the electrified time is shown in Figure 21.

It can be seen from Table 19 and Table 20 and Figure 21 that when the input voltage is 50 V and the room temperature is 15.3 °C, the temperature changes of the specimens under the two groups of conductive phase materials are 15.2 °C and 15.9 °C, respectively, and the average temperature increases by 7.60 °C and 7.95 °C per hour. The two groups of rut plate specimen temperature show very small change difference, have good heating effect, and meet the snow-melting ice requirements.

Through regression analysis, the linear fitting equations of the surface temperature T of the rutting plate specimen and the electrified time t are, respectively:Composite-doped 0.3% Carbon Fiber + 0.5 graphene: T = 0.1294t + 15.5501, R^2^ = 0.9971.0.3% carbon fiber + 0.6 graphene: T = 0.1314t + 15.9784, R^2^ = 0.9945(2)

It can be seen from the above equation that the correlation coefficients between the surface temperature of the specimen and the specimen under the two kinds of conductive phase materials are 0.9971 and 0.9945, respectively, indicating that the surface temperature of the specimen has a good correlation with the electrified time, and the process of CGA-10 electric energy to heat energy conversion is relatively stable.

The heating efficiency of the two groups of CGA-10 rutting plate specimens with different conductive phase content is shown in Table 21.

It can be seen from Table 21 that when the carbon fiber content is 0.3%, the input voltage is 50 V, and the continuous power is 120 min, the heating efficiency of the rutting plate specimens with two different graphene contents are 71.4% and 72.0%, respectively, and the difference in heating efficiency is small. When the graphene content is 0.5%, the rutting plate specimen has high heating efficiency, indicating that CGA-10 has less energy loss in the conversion process of electric energy and heat energy. When the external input power remains unchanged, the specimen exhibits good heating effect, indicating that the heat exchange and electrothermal conversion process with the external environment is stable.

### 6.5. Simulated Snow Melting Experiment

In order to study the actual ice-snow melting effect of 0.3% carbon fiber + 0.5 graphene and 0.3% carbon fiber + 0.6 graphene CGA-10, the low-temperature constant temperature test box was used to simulate the external low-temperature environment. The crushed ice residue was used instead of snow to spread evenly on the surface of the two groups of rutting plate specimens. The thickness of the crushed ice layer was 3 mm. The temperature of the low-temperature constant temperature test box was set to −10 °C. The temperature was maintained for 24 h to ensure that the rutting plate specimen was consistent with the ambient temperature. The input voltage was set to 50 V. The continuous power supply time was 10 h and the digital thermometer indicators were recorded every 30 min. Since the heat exchange between the object and the environment will be accelerated at low temperatures, to reduce the heat loss of the rutting plate specimen and ensure the accuracy of the test, the polystyrene foam plate is tightly wrapped at the bottom and around the specimen. The rutting board specimen melting snow test results are shown in Figure 22.

It can be seen from Figure 22 that:Under the condition that the input voltage of 50 V is continuously electrified for 600 min, the bottom temperature changes of the rutting plate specimens with 0.3% carbon fiber + 0.5 graphene and 0.3% carbon fiber + 0.6 graphene are 15.0 °C and 15.7 °C, respectively. The surface temperature changes of the specimens are 10.9 °C and 11.6 °C, respectively. The time required for the bottom temperature of the specimens to rise from −10 to 0 °C is 180 min, and the time required for the surface temperature to rise to 0 °C is 240 min and 230 min, respectively. The temperature changes in the low-temperature constant temperature test box are 0.8 °C and 0.9 °C, respectively, with little change.Compared with the heating rate of the rutting plate specimen at room temperature, the heating rate of the specimen surface during the ice-snow melting test was significantly slow, and the heating rate at the bottom of the specimen was significantly higher than that at the surface. This is because the heat exchange rate is accelerated and the heat loss is increased under the external low temperature environment, thus slowing the heating rate at the bottom of the specimen. At the same time, when the surface of the specimen reaches 0 °C, the heat generated continues to exchange with the outside world and store heat within itself. All the heat is involved in the process of melting ice and snow, and the surface of the specimen is no longer heated before the complete melting of ice and snow.The whole snow melting test process can be divided into three stages. The first stage is the heating process of ice and snow on the specimen and its surface, which is characterized by fast heating rate and stable heating effect. The second stage is the melting process of ice and snow on the surface of the specimen. The surface temperature of the specimen is basically maintained at 0 °C, and the heat transferred to the surface is basically absorbed by the melting of ice and snow. Stage 3 is the end of melting ice and snow, the specimen surface began to heat up, characterized by slow heating rate.

The calculation results of the energy conversion process in the whole snow melting process are shown in Table 22.

According to Table 22, the energy conversion relationship during snow melting can be obtained, as shown in Figure 23.

It can be seen from Table 22 and Figure 23 that:At 50 V input voltage, the input power of the power supply is only 26.59 W and 27.74 W with two kinds of conductive phase materials, which takes a long time to completely melt the ice and snow. Therefore, the time used in the ice-snow melting process can be shortened by increasing the input power and increasing the input power can be achieved by reducing the resistance of the specimen and increasing the voltage. Among them, increasing the input voltage threatens the safety of the test, and reducing the resistance provides higher safety.In the ice melting test, the heat generated by CGA-10 using the current thermal effect mainly includes the heat exchanged with low temperature environment (heat loss), the heat absorbed by ice melting and the heat saved by the specimen itself. The ice-snow melting efficiency of the two conductive phase materials was 28.73% and 29.81%, respectively. The heat storage energy of the specimen is 134.98 kJ and 141.99 kJ, respectively. The heat required for heating and melting of ice and snow is 109.85 kJ and 114.12 kJ, respectively. The heat loss was 138.06 kJ and 126.70 kJ, respectively, accounting for 36.1% and 33.1% of the total heat, respectively. There was little difference in snow melting efficiency between the two groups of rutting plate specimens, and more heat was transferred to the external environment in the form of heat conduction.By analyzing the energy conversion relationship, it can be found that heat loss accounts for a high proportion of the total heat, resulting in a decrease in snow melting efficiency. Therefore, to further improve the efficiency of ice and snow melting of CGA-10 and make more heat involved in the process of ice and snow melting, an insulation layer can be added below the conductive layer to prevent the downward transfer of heat and reduce heat loss, so that the heat energy transformed by conductive asphalt concrete can be effectively transferred to the road surface to ensure more efficient use of energy.

## 7. Conclusions

The mix design of GA-10 was optimized. The optimum content of Sasobit was determined to be 3.0% by flow test, and the optimum oil–stone ratio of GA-10 was determined to be 8.5% by penetration and penetration increment test. On this basis, the fluidity test, penetration, and penetration increment test were carried out with the addition of 0.3% carbon fiber + graphene (0.4%, 0.5%, 0.6%) three groups of content to determine that the optimum oil–stone ratio of CGA-10 was 9.8%. At this time, CGA-10 demonstrated good fluidity and plastic-deformation resistance.Compared with the specimen doped with 0.3% carbon fiber, the dynamic stability of CGA-10 doped with conductive phase materials increased by 12.3%, 17.6% and 22.9%, respectively, the flexural strength decreased by 5.4%, 10.0% and 16.4%, and the maximum flexural strain decreased by 5.8%, 8.0% and 9.9%, respectively. The results of the immersion Marshall test showed that the residual stability of CGA-10 Marshall specimen with conductive phase materials was reduced by 0.7%, 0.9% and 1.9%, respectively, compared with the specimen with 0.3% carbon fiber alone, with the increase of graphene content. The freeze–thaw splitting test results are consistent with the immersion Marshall test results. Overall, compared with GA-10, the water stability of CGA-10 with graphene and carbon fiber is better. The content of conductive phase materials was positively correlated with the fatigue resistance of CGA-10. Compared with the fatigue specimen doped with 0.3% carbon fiber alone, the average fatigue times of CGA-10 doped with conductive phase materials increased by 4.5%, 5.8% and 7.7%, respectively, with the increase of graphene content.It is recommended that the optimum graphite distribution between the specimen and the electrode contact surface is 600 g/m^2^, and the contact resistance is the smallest. When the conductive phase material content is 0.3% carbon fiber + 0.4% graphene, the resistivity of the specimen is the largest, which is 6.81 Ω·m. When the graphene content increases from 0.4% to 0.5%, the resistivity of the specimen decreases by 2.11 Ω·m. When the graphene content continues to increase, the resistivity of the specimen changes slightly, and the conductivity of CGA-10 has reached the percolation threshold. When the conductive phase material is 0.3% carbon fiber + 0.5% graphene and 0.3% carbon fiber + 0.6% graphene, the resistivity of rut plate specimen is 4.70 Ω·m and 4.51 Ω·m, respectively, which has good electrical conductivity.Based on the SEM microscopic analysis, it is found that graphene is a multilayer sheet stacking structure, and there are many stripes on the surface of carbon fiber. The carbon fiber overlapped with each other is well combined and wrapped with minerals and asphalt. Due to the large aspect ratio of carbon fiber, it can play the role of a conductive bridge in CGA-10, connect graphene uniformly distributed in asphalt mortar, and construct a complete conductive network.The electrified heating test showed that when 0.3% carbon fiber + 0.5% graphene and 0.3% carbon fiber + 0.6% graphene were mixed, the correlation coefficients between the surface temperature of the rutting plate specimen and the electrified time were 0.9971 and 0.9945, respectively. The two showed good correlation, and the heating efficiency of the specimen was 71.4% and 72.0%, respectively, with high electrothermal conversion efficiency.When 0.3% carbon fiber + 0.5% graphene and 0.3% carbon fiber + 0.6% graphene are mixed, the heat loss of rut plate specimens account for 36.1% and 33.1% of the total heat, respectively. The heat insulation layer can be added below the conductive layer to prevent downward heat transfer and reduce heat loss. The snow-melting efficiency of the two groups of rut plate specimens is 28.73% and 29.81%, respectively, and the difference is small. However, the low-temperature crack resistance and water stability of CGA-10 will decrease with an increase in graphene content. Therefore, we recommend that the best content of conductive phase material in CGA-10 is 0.3% carbon fiber + 0.5% graphene.

## Figures and Tables

**Figure 1 polymers-15-01864-f001:**
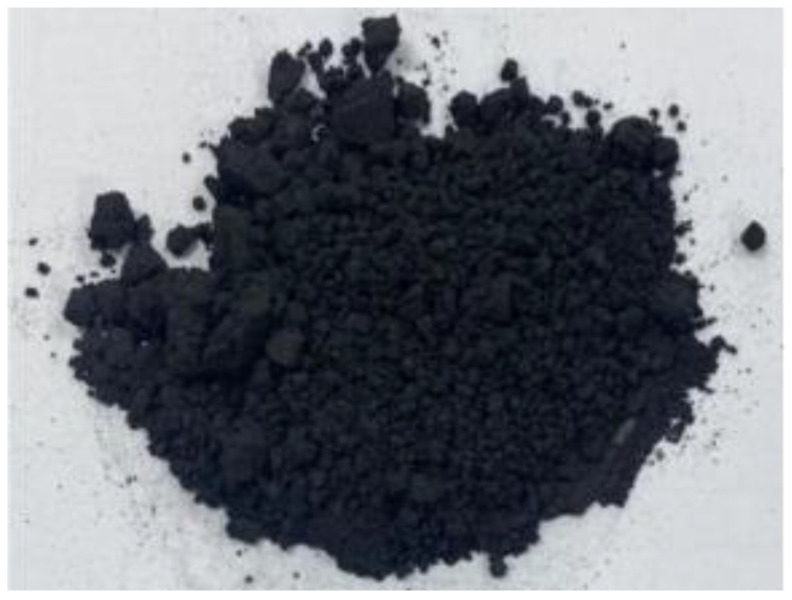
Graphene.

**Figure 2 polymers-15-01864-f002:**
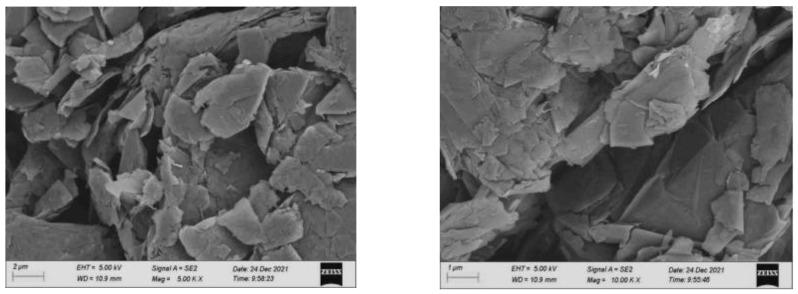
Micromorphology of Graphene. (**a**) 5000 times. (**b**) 10,000 times.

**Figure 3 polymers-15-01864-f003:**
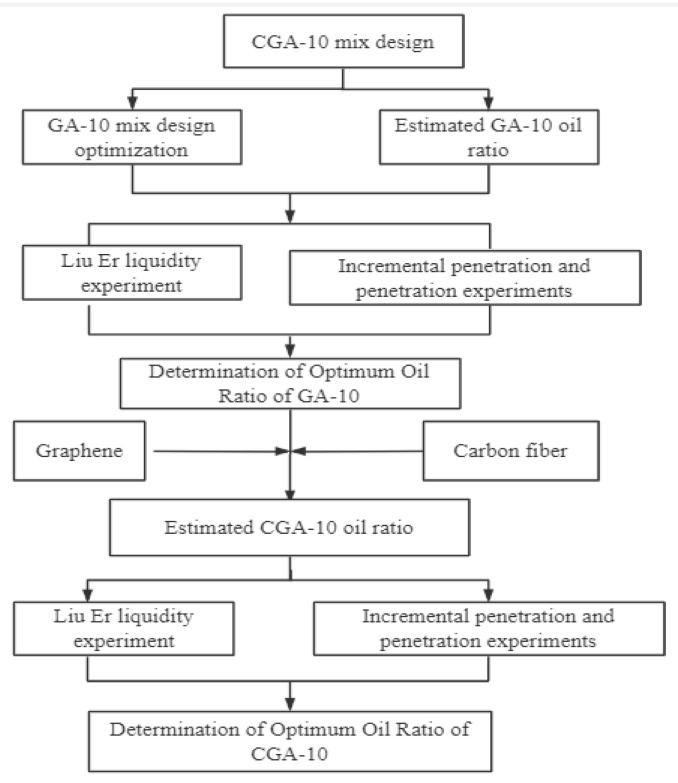
Mix Proportion Design of Casting Conductive Asphalt Concrete.

**Figure 4 polymers-15-01864-f004:**
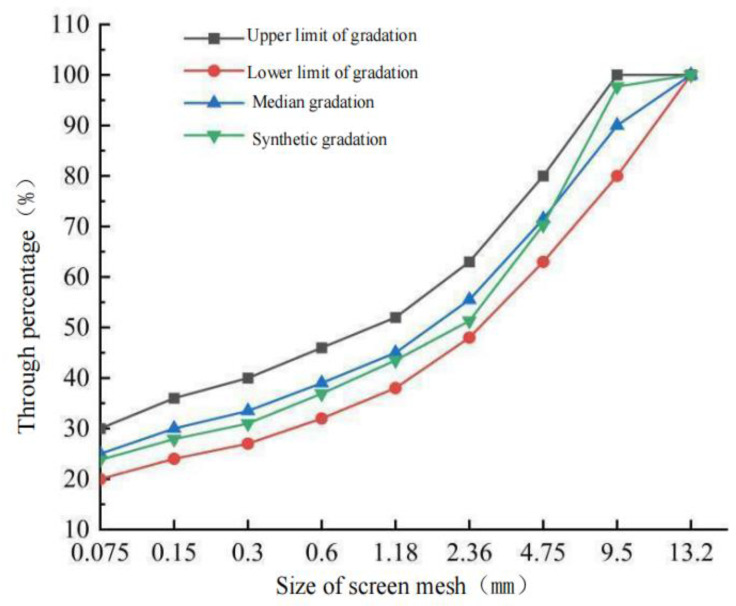
Aggregate gradation of GA-10.

**Figure 5 polymers-15-01864-f005:**
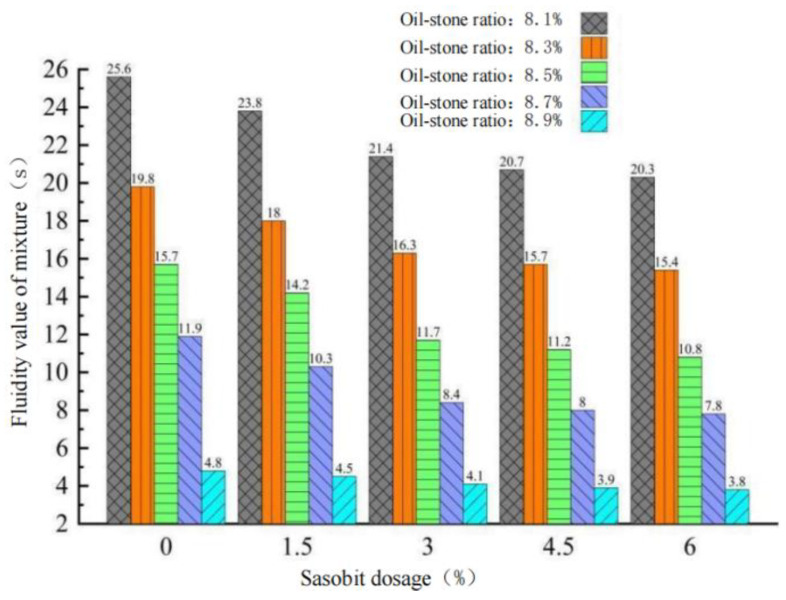
Effect of Sasobit content on fluidity of GA-10 asphalt mixture.

**Figure 6 polymers-15-01864-f006:**
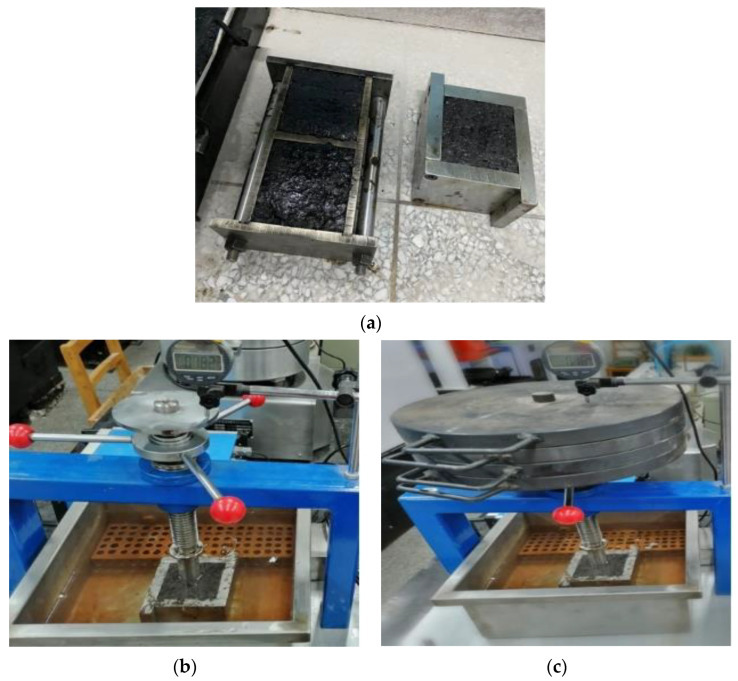
Penetration and penetration increment test of GA-10. (**a**) GA-10 Penetration specimen, (**b**) Initial pressure of penetration specimen, (**c**) Final pressure of penetration specimen.

**Figure 7 polymers-15-01864-f007:**
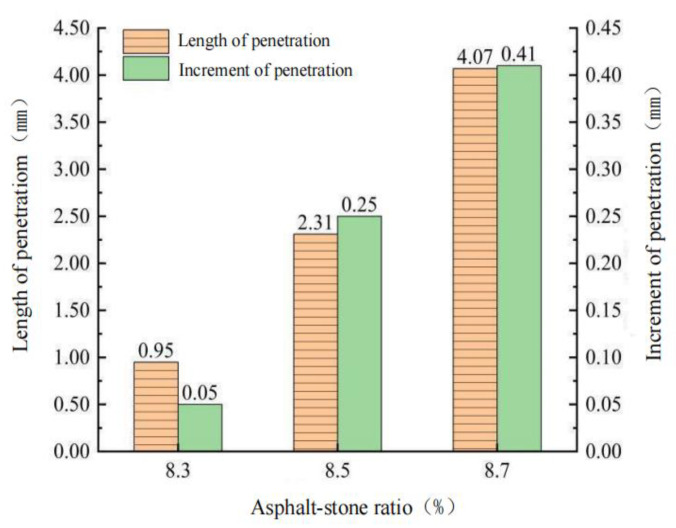
Test results of penetration and penetration increment with different oil–stone ratio.

**Figure 8 polymers-15-01864-f008:**
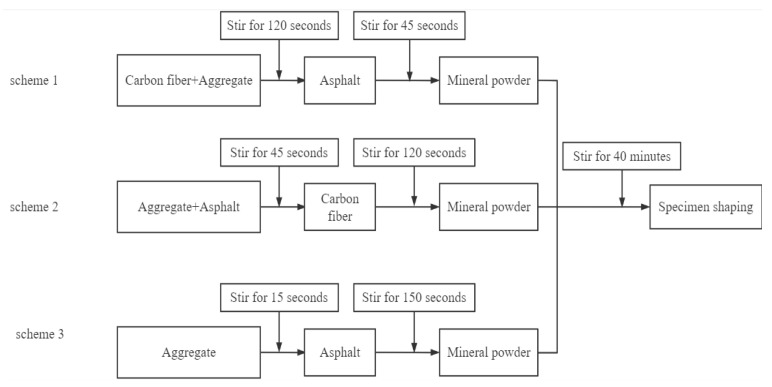
Carbon fiber dispersion scheme.

**Figure 9 polymers-15-01864-f009:**
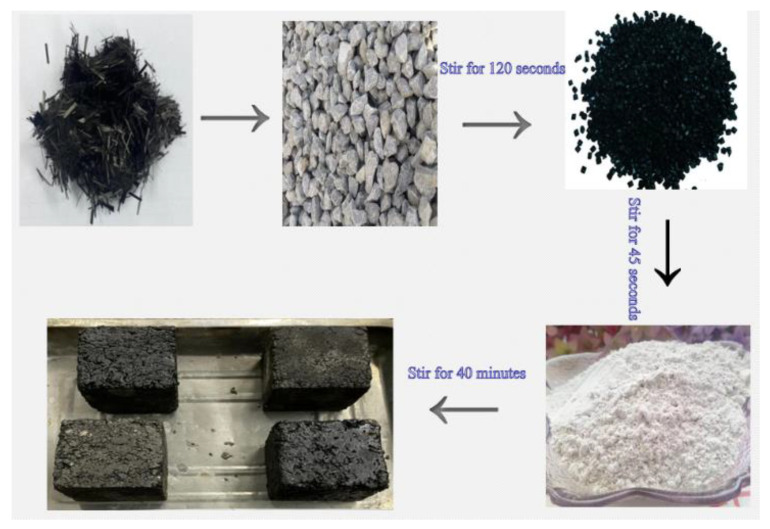
The experimental process.

**Figure 10 polymers-15-01864-f010:**
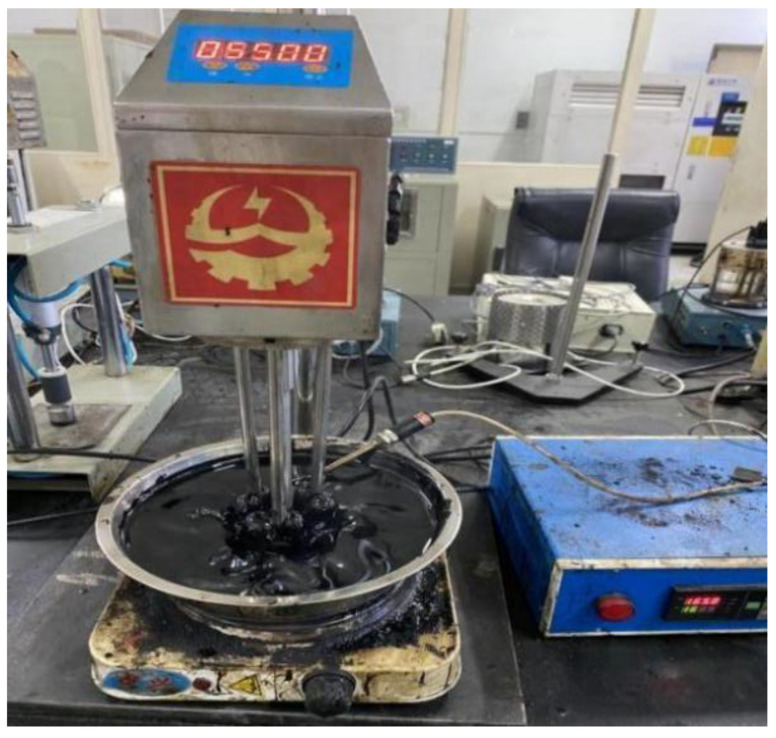
Graphene dispersion process.

**Figure 11 polymers-15-01864-f011:**
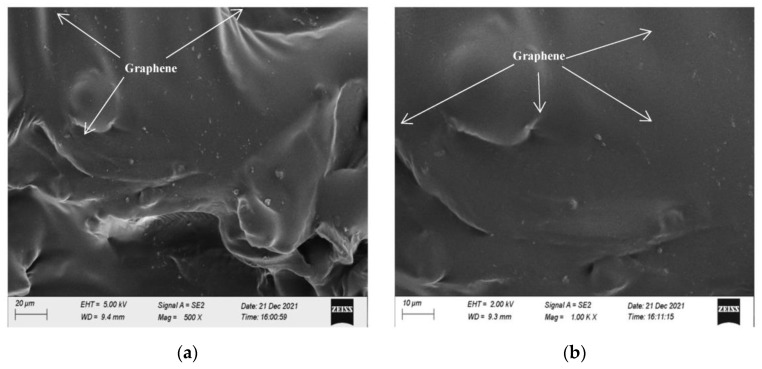
Microstructure of dispersed graphene asphalt. (**a**) 500 times, (**b**) 1000 times.

**Figure 12 polymers-15-01864-f012:**
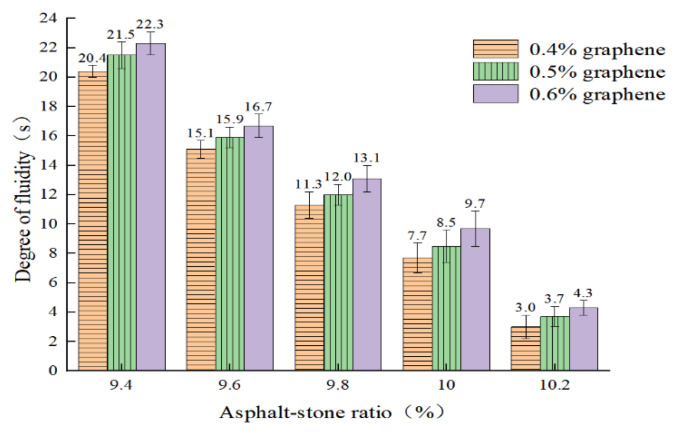
Fluidity test of CGA-10 asphalt mixture with 0.3% carbon fiber content.

**Figure 13 polymers-15-01864-f013:**
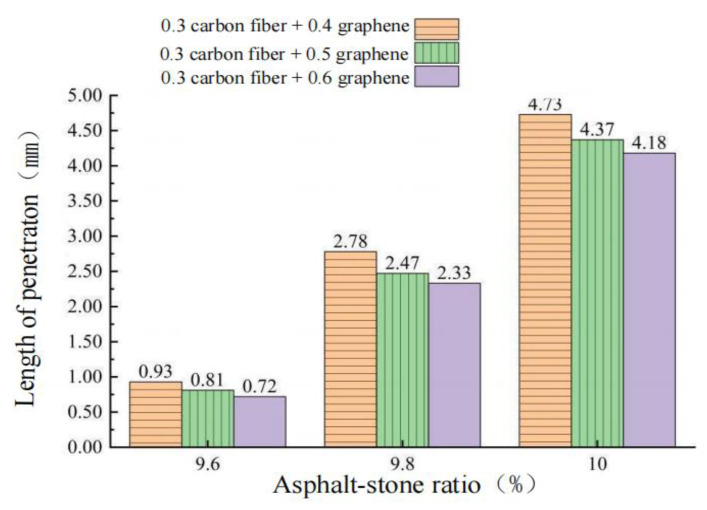
Test results of penetration with different oil–stone ratio.

**Figure 14 polymers-15-01864-f014:**
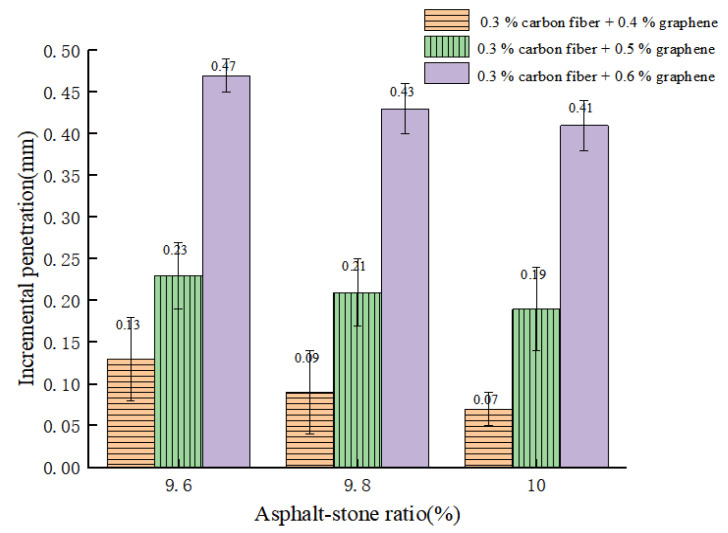
Test results of penetration increment with different oil–stone ratio.

**Figure 15 polymers-15-01864-f015:**
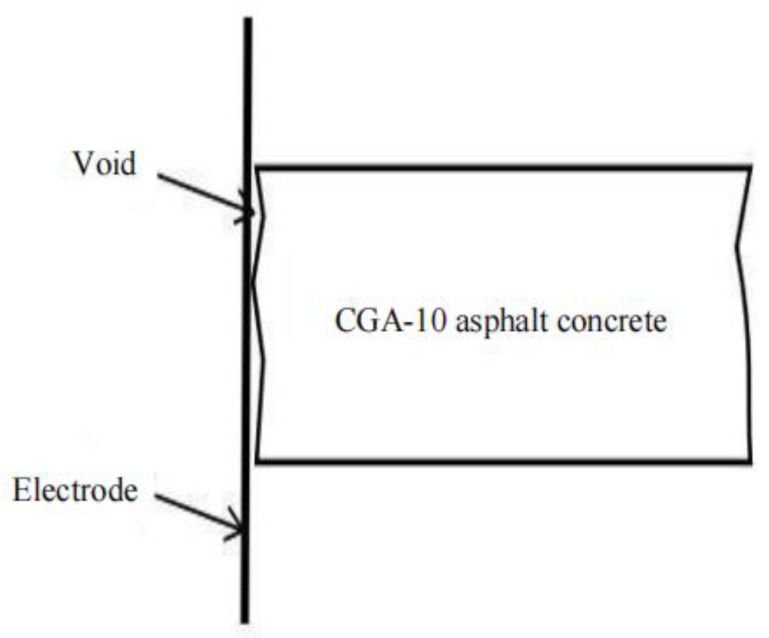
Contact surface between CGA-10 specimen and electrode sheet.

**Figure 16 polymers-15-01864-f016:**
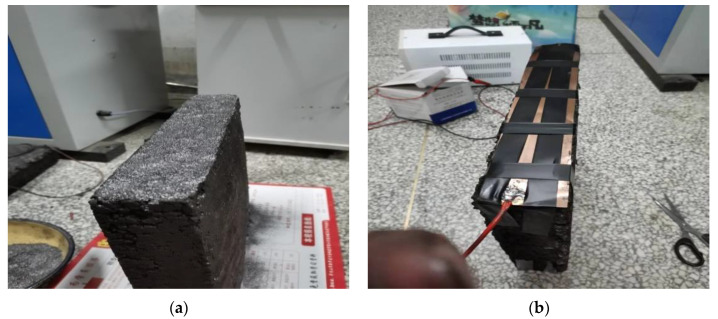
The contact surface of the specimen is covered with graphite. (**a**) Graphite spread, (**b**) Electrode bonding.

**Figure 17 polymers-15-01864-f017:**
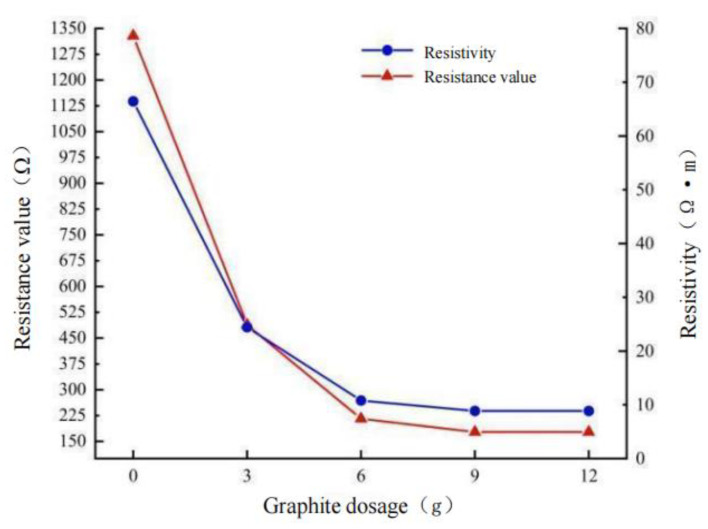
Diagram showing resistance variation trend of specimen.

**Figure 18 polymers-15-01864-f018:**
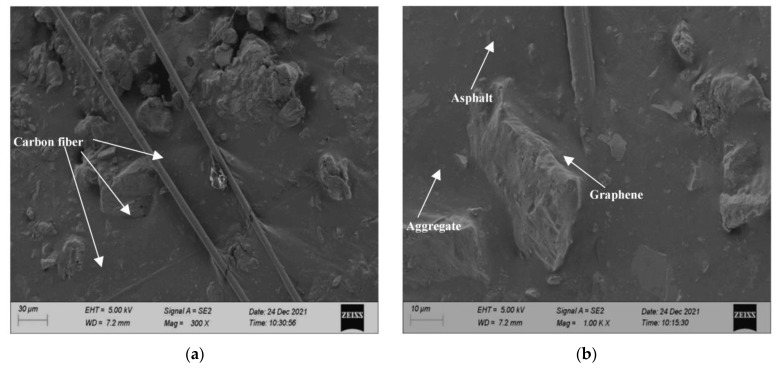
Microstructure of CGA-10. (**a**) 300 times, (**b**) 1000 times.

**Figure 19 polymers-15-01864-f019:**
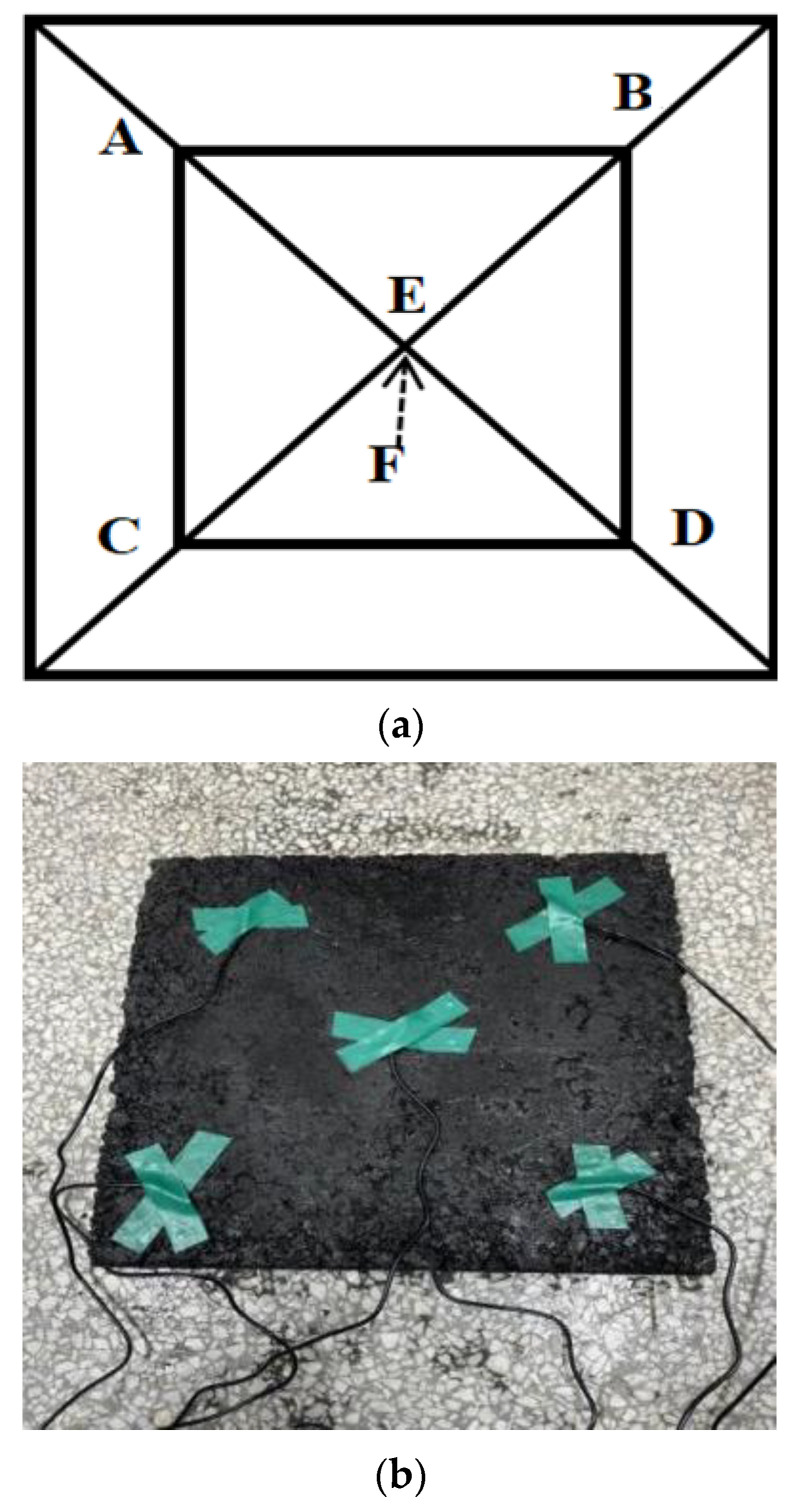
Temperature measuring point distribution of rut plate specimen in temperature rise test. (**a**) Diagram of distribution of temperature measuring points, (**b**) Temperature measuring point distribution of rutting plate.

**Figure 20 polymers-15-01864-f020:**
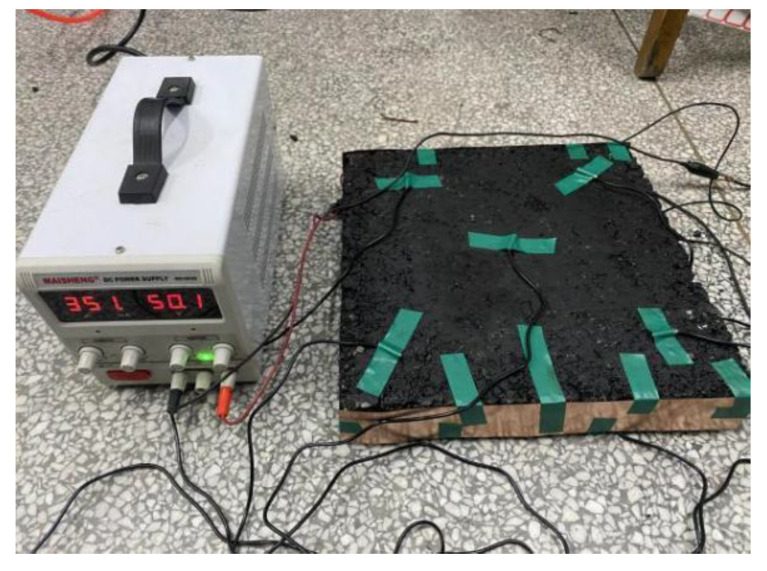
Electric heating test of rut plate specimen.

**Figure 21 polymers-15-01864-f021:**
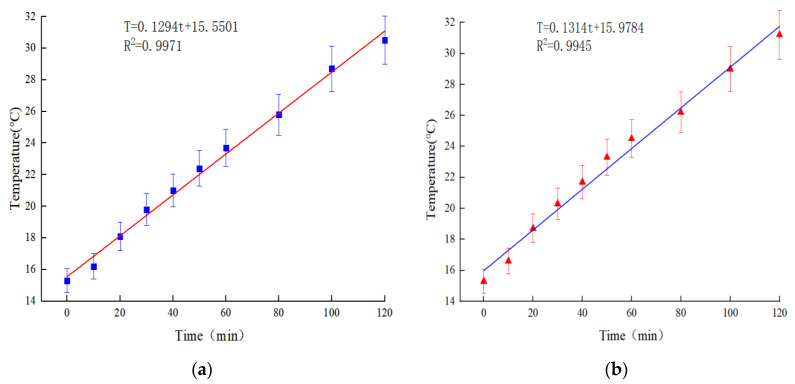
Temperature and electrification time of rut plate specimen. (**a**) 0.3% carbon fiber + 0.5% graphene. (**b**) 0.3% carbon fiber + 0.6% graphene.

**Figure 22 polymers-15-01864-f022:**
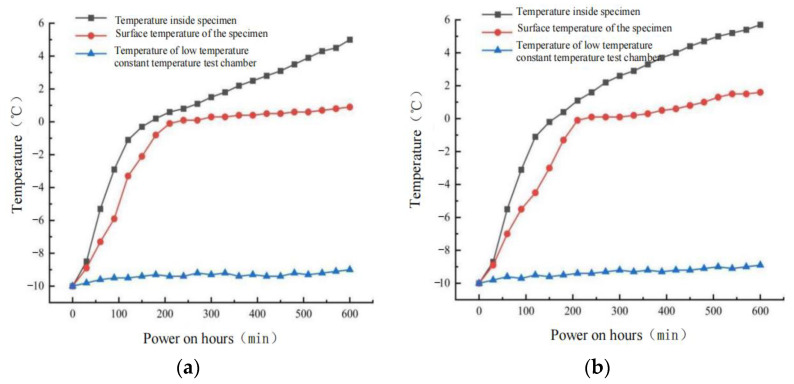
Melting ice and snow test results under different dosage. (**a**) 0.3% carbon fiber + 0.5% graphene content. (**b**) 0.3% carbon fiber + 0.6% graphene.

**Figure 23 polymers-15-01864-f023:**
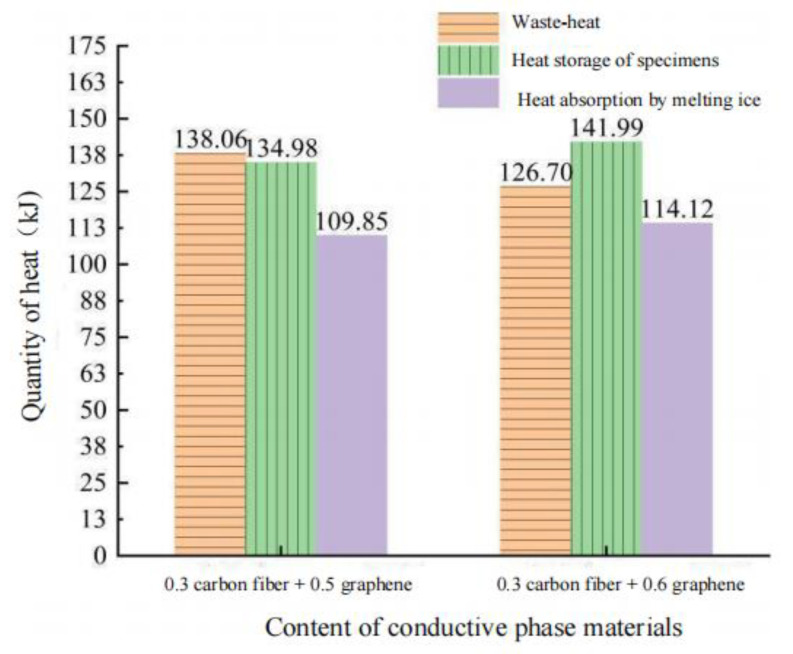
Energy transformation of rut board specimen during melting ice and snow.

**Table 1 polymers-15-01864-t001:** Main technical indexes of SBS-modified asphalt.

Targets of Test	Penetration Degree (25 °C, 100 g, 5 s) (10^−1^ mm)	Softening Point (Global Method) (°C)	Ductility (5 cm/min, 10 °C) (cm)	Flash Point (°C)	Density (g/cm^3^)	Wax Content (%)	Solubility (%)	After RTFOT (%)		Ductility (5 cm/min, 10 °C) (cm)
								Mass Variation	Penetration Ratio	
Test result	54.1	77.1	33	674	1.137	1.3	99.5	-0.3	77	16
Technical requirement	40~60	≥60	≥20	≥230	actual	≤3.0	≥99.0	≤±1.0	≥65	≥15

**Table 2 polymers-15-01864-t002:** Main technical indexes of Sasobit.

Targets of Test	Appearance	Chemical Composition	Drop Melting Point (°C)	Flash Point (°C)
Measured value	White solid small particles	Long-chain aliphatic alkanes	110.0	285.0

**Table 3 polymers-15-01864-t003:** Main technical indexes of 3~5 mm gravel.

Test Index	Apparent Specific Gravity (g/cm^3^)	Relative Density of Bulk Volume (g/cm^3^)	Water Absorption (%)	Content of Needle Flake Particles in Mixture (%)	Water Washing Method < 0.075 mm Particle Content (%)	Soft Stone Content (%)
Test result	2.718	2.643	0.54	12.5	0.3	0.2
Technical requirement	≥2.60	actual	≤2.0	≤15	≤1	≤3

**Table 4 polymers-15-01864-t004:** Main technical indexes of 5~10 mm gravel.

Test Index	Losangeles Weared Value (%)	Apparent Specific Gravity (g/cm^3^)	Relative Density of Bulk Volume (g/cm^3^)	Water Absorption(%)	Content of Needle Flake Particles in Mixture (%)	Water Washing Method < 0.075 mm Particle Content (%)	Soft Stone Content (%)
Test result	18.1	2.723	2.645	0.52	13.1	0.5	0.4
Technical requirement	≤28	≥2.60	actual	≤2.0	≤15	≤1	≤3

**Table 5 polymers-15-01864-t005:** Main technical indexes of fine aggregate.

Test Index	Apparent Specific Gravity (g/cm^3^)	Sand Equivalent (%)	Angularity (%)
Test result	2.714	72.6	33.2
Technical requirement	≥2.50	≥60	≥30

**Table 6 polymers-15-01864-t006:** Main technical indexes of mineral powder.

Targets of Test	Apparent Density (t/m^3^)	Water Content (%)	Size Range <0.6 mm (%)	<0.15 mm (%)	<0.075 mm (%)	Hydrophilic Coefficient
Test result	2.726	0.2	100	96.8	86.5	0.50
Technical requirement	≥2.50	≤1	100	90–100	75–100	<1.0

**Table 7 polymers-15-01864-t007:** Technical specifications of graphene.

Performance Parameter	Appearance	Purity (wt%)	Thickness (nm)	Specific Surface Area (m^2^/g)	Layer Diameter (μm)
Actual	black powder	95	6	89	47
Technical requirement	black powder	>90	3~8	50~150	10~50

**Table 8 polymers-15-01864-t008:** Technical index of carbon fiber.

Performance Parameter	Tensile Modulus (GPa)	Density (g/cm^3^)	Tensile Strength (MPa)	Fiber Diameter (μm)	Bulk Density (g/cm^3^)	Carbon Content (%)
Actual	220	1.76	3800	7	0.43	98.35
Technology index	>200	>1.72	>3500	6~8	>0.4	>97

**Table 9 polymers-15-01864-t009:** Screening results of various aggregates and powders.

Aggregate	Percentage of Passing through Each Sieve Hole (%)								
	13.2	9.5	4.75	2.36	1.18	0.6	0.3	0.15	0.075
1#	100	91.6	1.3	0.4	0.4	0.3	0.4	0.4	0.3
2#	100	100.0	86.0	4.8	1.2	1.0	1.0	1.0	1.0
3#	100	100.0	100.0	96.5	69.7	44.8	22.8	13.3	8.7
Mineral fines	100	100.0	100.0	100.0	100.0	100.0	99.1	96.4	85.0

**Table 10 polymers-15-01864-t010:** Aggregate gradation of GA-10.

Sieve Pore Diameter (mm)	13.2	9.45	4.75	2.36	1.18	0.6	0.3	0.15	0.075
Upper limit of gradation	100	100	80	63	52	46	40	36	30
Lower limit of gradation	100	80	63	48	38	32	27	24	20
Median gradation	100	90	71.5	55.5	45	39	33.5	30	25
Synthetic gradation	100.0	97.7	70.3	51.3	43.5	36.9	31.0	27.9	23.8

**Table 11 polymers-15-01864-t011:** GA-10 mix design inspection.

Mixture Type	Flexural Tensile Strength (MPa)	Maximum Bending Strength (με)	Bending Stiffness Modulus (MPa)
GA-10	10.073	3878.7	2597.0
Technology index	—	3000	—

**Table 12 polymers-15-01864-t012:** Results of rutting test for asphalt mixture.

Content of Conductive Phase Materials	45 min Deformation (mm)	60 min Deformation (mm)	Dynamic Stability (times/mm)
0.3% Carbon fiber + 0.4% graphene	1.90	2.23	1885
0.3% Carbon fiber + 0.5% graphene	1.76	2.08	1973
0.3% Carbon fiber + 0.6% graphene	1.67	1.98	2063
0.3% Carbon fiber	1.99	2.37	1678
Undoped (GA-10)	2.29	2.72	1473
Technology index	—	—	300

**Table 13 polymers-15-01864-t013:** Results of low-temperature bending test of asphalt mixture.

Content of Conductive Phase Materials	Flexural Strength (MPa)	Maximum Bending Strength (με)	Bending Stiffness Modulus (MPa)
0.3% Carbon fiber + 0.4% graphene	12.958	4577.7	2800.7
0.3% Carbon fiber + 0.5% graphene	12.335	4470.3	2759.3
0.3% Carbon fiber + 0.6% graphene	11.773	4378.1	2689.1
0.3% Carbon fiber	13.701	4857.8	2820.4
Undoped (GA-10)	10.022	3818.0	2624.9
Technology index	—	3000	—
Standard deviation	1.246	341.534	78.464

**Table 14 polymers-15-01864-t014:** Marshall test results of asphalt mixture immersion.

Content of Conductive Phase Materials	30 min Immersion Stability (kN)	48h Stability of Immersion (kN)	Residual Stability (%)
0.3% Carbon fiber + 0.4% graphene	13.09	12.17	93.0
0.3% Carbon fiber + 0.5% graphene	12.45	11.53	92.6
0.3% Carbon fiber + 0.6% graphene	11.83	10.86	91.8
0.3% Carbon fiber	13.75	12.88	93.7
Undoped (GA-10)	10.78	9.54	88.5
Technology index	—	—	≥80

**Table 15 polymers-15-01864-t015:** Test results of freezing–thawing splitting of asphalt mixture.

Content of Conductive Phase Materials	Splitting Strength before Freezing and Thawing (MPa)	Splitting Strength after Freezing and Thawing (MPa)	TSR (%)
0.3% Carbon fiber + 0.4% graphene	1.217	1.117	91.8
0.3% Carbon fiber + 0.5% graphene	1.193	1.087	91.1
0.3% Carbon fiber + 0.6% graphene	1.172	1.058	90.3
0.3% Carbon fiber	1.383	1.195	92.8
Undoped (GA-10)	0.981	0.848	86.4
Technology index	—	—	≥80

**Table 16 polymers-15-01864-t016:** Fatigue test results of asphalt mixture.

Content of Conductive Phase Materials	Average Fatigue Times (times)
0.3% Carbon fiber + 0.4% graphene	349,974
0.3% Carbon fiber + 0.5% graphene	354,226
0.3% Carbon fiber + 0.6% graphene	360,533
0.3% Carbon fiber	334,787
Undoped (GA-10)	315,375
Technology index	—
Standard deviation	16,202.825

**Table 17 polymers-15-01864-t017:** Resistance test results of externally attached copper electrode rut plate specimen.

Content of Conductive Phase Materials	Specimen Height H (mm)	Resistance R(Ω)	Average Resistance R(Ω)	Resistivity *ρ* (Ω·m)
0.3% Carbon fiber + 0.4% graphene	50.1	135.3	136.2	6.81
	50.3	136.9		
	50.3	136.2		
0.3% Carbon fiber + 0.5% graphene	50.1	95.3	94.0	4.70
	50.3	93.7		
	50.3	95.0		
0.3% Carbon fiber + 0.6% graphene	50.2	89.8	90.1	4.51
	50.1	91.2		
	50.2	90.3		

**Table 18 polymers-15-01864-t018:** Electrothermal parameters of some materials related to the test.

Name of Material	Specific Heat Capacity J/(kg·K)	Density (kg/m^3^)
Ice	2050.0	917
Modified bituminous concrete	813.2	2415
Carbon fiber	720.0	1760
Graphene	710.0	1800

Note: The heat of dissolution of ice is 3.35 × 105 (J/kg).

**Table 19 polymers-15-01864-t019:** Temperature rises test results of specimens with 0.3% carbon fiber + 0.5% graphene content.

**Time (min)**	0	10	20	30	40	50	60	80	100	120
**Temperature (°C)**	15.3	16.2	18.1	19.8	21.0	22.4	23.7	25.8	28.7	30.5

**Table 20 polymers-15-01864-t020:** Temperature rises test results of specimens with 0.3% carbon fiber + 0.6% graphene content.

**Time (min)**	0	10	20	30	40	50	60	80	100	120
**Temperature (°C)**	15.3	16.6	18.7	20.3	21.7	23.3	24.5	26.2	29.0	31.2

**Table 21 polymers-15-01864-t021:** Results of temperature rise and heat test of rut plate specimen.

Content of Conductive Phase Materials	Resistance (Ω)	Resistivity (Ω·m)	Input Power (W)	Total Heat (kJ)	Specimen Specific Heat J/(kg·K)	Heating-Up (°C)	Heat Storage of Specimen (kJ)	Heat Generation Efficiency (%)
0.3% Carbon fiber + 0.5% graphene	94.0	6.64	26.59	191.45	820.3	15.2	136.78	71.4
0.3% Carbon fiber + 0.6% graphene	90.1	6.36	27.74	199.73	827.5	15.9	143.81	72.0

**Table 22 polymers-15-01864-t022:** Energy calculation of snow and ice melting process simulated by rut plate specimen.

Parameter	Sign	Unit	Content of Conductive Phase Materials	
			0.3% Carbon Fiber + 0.5% Graphene	0.3% Carbon Fiber + 0.6% Graphene
Resistance	R	Ω	94.0	90.1
Power on hours	t	s	14,400	13,800
Input power	P = U^2^/R	w	26.59	27.74
Total heat	Q = Pt	kJ	382.89	382.81
Specimen mass	M_1_	kg	10.97	10.93
Specimen specific heat	C_1_	J/(kg·K)	820.3	827.5
Ice quality	M_2_	kg	0.309	0.321
Ice specific heat	C_2_	J/(kg·K)	2050.0	2050.0
the solution heat of ice	Q_i_	J/(kg·K)	3.35 × 10^5^	3.35 × 10^5^
Temperature rises of specimen	T_1_	°C	15.0	15.7
Ice rising temperature	T_2_	°C	10	10
Heat storage energy of specimen	Q_1_ = M_1_C_1_T_1_	kJ	134.98	141.99
Heat uptake by Ice	Q_2_ = M_2_C_2_T_2_	kJ	6.33	6.58
Complete melting of ice absorbs heat	Q_3_ = M_2_Q_i_	kJ	103.52	107.54
Ice melting efficiency	H = (Q_2_ + Q_3_)/Q	%	28.73%	29.81%

## Data Availability

All data that support the findings of this study are included within the article.

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
