# Peer review of "Preparation and Properties of Composite Graphene/Carbon Fiber Pouring Conductive Asphalt Concrete"

_polymers, 2023, doi:10.3390/polym15081864_

Round 1

Reviewer 1 Report

Journal Name: Polymers

Title: Preparation and Properties of Composite Graphene /Carbon Fiber Pouring Conductive Asphalt Concrete

In the current article Y. Chen et al., graphene composite to solve the snow problem on a steel bridge. The results are convincing but several articles are already published related to the GNP-PANI system this is the main concern, therefore, authors have to improve the literature review and discuss the gap. I recommend a major revision to publish at MDPI Polymers.  

Major revision

1.      The abstract should be more concise

2.      English has to be improved

3.      Dear authors please have a look at the following articles and discuss how the current work is different from the previous publications

https://ascelibrary.org/doi/full/10.1061/%28ASCE%29MT.1943-5533.0004183

https://www.sciencedirect.com/science/article/pii/S0950061821022686

4.      Graphical abstract may be useful

5.      By SEM analysis the authors must discuss the flake properties of graphene

6.      Resolutions of the images should be improved

7.      Error bars of Fig. 8 and 10 must be mentioned

8.      Discuss further increase in resilient modulus for the different graphene mixture concentrations.

9.      Prepare a scheme to find the optimum asphalt dosage it will be helpful for general readers.

10.  Error bars of Fig. 17 (Discuss further the importance of R2 values)

Reviewer 3 Report

The article presents a study on the use of conductive gussasphalt concrete (CGA) to address the problem of snow accumulation on steel bridge areas. The authors investigated the performance of CGA prepared by adding conductive phase materials, such as graphene and carbon fiber, to gussasphalt. They conducted several tests to evaluate the design mix and the influence of different conductive phase material content on the CGA's conductivity and microstructure characteristics.

Overall, the study is well-conducted, and the authors have taken a systematic approach to studying the performance of CGA with different conductive phase materials. However, I believe that the manuscript requires significant revisions before it is ready for publication.

1       I recommend the addition of a dedicated section outlining the experimental procedures in more detail. This would help readers to better understand the study's methodology and results, and could include information about test conditions and measurement techniques, as well as any other relevant details that could help readers replicate the experiments.

2    2) I noticed that the tables in the manuscript are not consistently formatted. For instance, some tables include units as separate columns (e.g., Table 1), while others include units in the column names (e.g., Table 11). I recommend that the authors ensure all tables in the manuscript are formatted consistently, preferably following the template used in Table 11, where the column names include the units.

3    3)  I have some concerns regarding the use of graphite powder to improve electrical contact, as it could potentially affect the accuracy or precision of the conductivity measurements. I recommend that the authors consider alternative methods, such as polishing the surface of the samples or using a different type of electrode. The manuscript could also benefit from providing more detail on the methodology used to measure electrical conductivity, including whether similar methods have been used in other studies and any potential sources of error or uncertainty associated with the use of graphite powder.

4 4)      Furthermore, while Figure 17 provides valuable information on the temperature rise of the CGA samples during the heating test, I suggest that the authors include additional data on the temperature behavior of all five thermocouples. This would allow readers to more fully understand the temperature behavior of the CGA samples over time.

55)      Lastly, I recommend that the authors combine Figures 18 and 19 into a single figure to improve the organization and clarity of the results presentation. Placing the two figures side by side in a single panel, with sub-panels (a) and (b) corresponding to the data shown in Figures 18 and 19, respectively, would allow readers to more easily compare and contrast the results from the two experiments, and would help to avoid redundancy and confusion in the figures.

Overall, these revisions would significantly enhance the manuscript's overall clarity and readability, and I strongly encourage the authors to consider these suggestions.

Round 2

Reviewer 1 Report

The article can be accepted in its current form

Reviewer 3 Report

I recommend the publication of the article "Preparation and Properties of Composite Graphene /Carbon Fiber Pouring Conductive Asphalt Concrete". The authors have addressed all my previous concerns and have made significant improvements to the manuscript.